# In Vitro Analysis of the Antagonistic Biological and Chemical Interactions between the Endophyte *Sordaria tomento-alba* and the Phytopathogen *Botrytis cinerea*

**DOI:** 10.3390/ijms25021022

**Published:** 2024-01-13

**Authors:** Hernando José Bolívar-Anillo, Inmaculada Izquierdo-Bueno, Estrella González-Rey, Victoria E. González-Rodríguez, Jesús M. Cantoral, Isidro G. Collado, Carlos Garrido

**Affiliations:** 1Departamento de Química Orgánica, Facultad de Ciencias, Universidad de Cádiz, Puerto Real, 11510 Cadiz, Spain; hernando.bolivar@unisimon.edu.co (H.J.B.-A.); inmaculada.izquierdo@uca.es (I.I.-B.); estrella.gonzalezrey@alum.uca.es (E.G.-R.); 2Programa de Microbiología, Facultad de Ciencias Básicas y Biomédicas, Universidad Simón Bolívar, Barranquilla 080002, Colombia; 3Departamento de Biomedicina, Biotecnología y Salud Pública, Área de Microbiología, Facultad de Ciencias del Mar y Ambientales, Universidad de Cádiz, Puerto Real, 11510 Cadiz, Spain; victoriaeugenia.gonzalez@uca.es (V.E.G.-R.); jesusmanuel.cantoral@uca.es (J.M.C.)

**Keywords:** *Botrytis cinerea*, *Sordaria tomento-alba*, biocontrol, *Gliricidia sepium*, endophytic fungus, metabolites

## Abstract

Plant pathogenic infections causing substantial global food losses are a persistent challenge. This study investigates a potential biocontrol strategy against the necrotrophic fungus *Botrytis cinerea* using the endophytic fungus *Sordaria tomento-alba* isolated from *Gliricidia sepium* in Colombia. Today, synthetic fungicides dominate *B. cinerea* control, raising environmental and health concerns. *S. tomento-alba* exhibits notable in vitro effects, inhibiting *B. cinerea* growth by approximately 60% during co-culture and 50% in double disc co-culture. Additionally, it suppresses botryanes production and produces the compound heptacyclosordariolone, which has proven effective in inhibiting *B. cinerea* mycelial growth and spore germination in vitro. This biocontrol agent could be a potential eco-friendly alternative to replace synthetic fungicides. Our study provides insights into the chemical and biological mechanisms underpinning the antagonistic activity of *S. tomento-alba*, emphasizing the need for further research to understand its biosynthesis pathways and optimize its biocontrol potential. It also contributes molecular evidence of fungal interactions with implications for advanced forums in molecular studies in biology and chemistry, particularly in addressing plant pathogenic infections and promoting sustainable agriculture.

## 1. Introduction

*Botrytis* is a highly diverse genus, comprising multiple species with variations in biology, ecology, morphology, and hosts [1]. It is a plant pathogen affecting 586 host plant genera of agricultural and horticultural interest [2,3,4]. Moreover, *Botrytis* species have adapted to a wide range of geographical regions spanning from extremely cold to desert areas [4]. Among the different species, *Botrytis cinerea*, the cause of gray mold, is regarded as the most significant due to the incalculable economic losses it causes (estimated at between EUR 10 billion–100 billion annually). This is largely attributed to its extensive host range, which includes approximately 1400 species of ornamental and agricultural plants [4,5,6,7]. *B. cinerea*’s negative economic impact on crops has positioned it as one of the most extensively studied necrotrophic pathogens and the second most important plant pathogen globally [5].

During its life cycle, *B. cinerea* can establish three types of interactions with plants: pathogenic infection, latent infection, and an endophytic relationship [5,8]. Each of these stages is determined by conditions such as the physiological state of the plant, the environmental conditions at a specific moment in time, and interactions with the plant microbiota, among others [9]. Regarding pathogenic infection, *B. cinerea* is able to infect plants thanks to a wide arsenal of virulence factors such as secretion of enzymes (e.g., cutinase, lipase, and endopolygalacturonase), reactive oxygen species, and phytotoxins [10,11]. The main phytotoxins of *B. cinerea* are grouped into two families, the botryanes (terpenes) and the botcinins (polyketides) [12]. Botryanes include botrydial, dihydrobotrydial, and other derivatives with the same botryiane skeleton, while botcinins include, among others, botcinic acid and botcinins A and B [12].

Botrydial, a non-host specific phytotoxin, plays a fundamental role in the infection process of this fungus, enabling it to kill the cells of host plants solely through the action of this phytotoxin [12]. In addition, it has been shown that *B. cinerea* can increase botrydial production in the presence of other microorganisms as a possible defense mechanism, leading to the belief that this phytotoxin also has antimicrobial properties [13,14].

Currently, *B. cinerea* is mainly controlled by means of chemical fungicides, which have a negative impact on human and animal health (e.g., plant pollinators) and the environment [15]. Chemical fungicides can also have negative effects on plant physiology by altering photosynthesis and the composition of the microbial communities associated with them [16,17,18]. Despite these drawbacks, it is estimated that EUR 1 billion is still spent globally on the chemical control of gray mold annually [5]. In recent years, increasingly strict regulations have been established worldwide to control the use of chemical pesticides in agriculture (e.g., Directive 2009/128/EC and Regulation (EC) No 1107/2009), restricting the number of active ingredients available on the market for the control of phytopathogens [17,19,20]. Moreover, *B. cinerea* is considered a high-risk fungus considering its ability to build up resistance to chemical fungicides and thus limiting their efficacy. For all these reasons, it is vitally important to search for new alternatives for the efficient and rational control of this phytopathogen [21].

The use of microorganisms and/or their metabolites to control plant disease (biocontrol) is receiving a great deal of attention because this strategy is environmentally friendly and has no negative impact on human or animal health [22,23]. Microorganisms serving as biocontrol agents can inhibit phytopathogens directly (physical contact or specific mechanisms against the pathogen) or indirectly (stimulation of plant defenses, competition for substrates, etc.) or by mixed antagonism (antibiotics, lytic enzymes, etc.), all of which are compatible and can act simultaneously or synergistically [19,24,25]. The microorganisms that have been used in biocontrol to date are mainly isolated from plant environments such as the rhizosphere, endosphere, phyllosphere, and spermosphere [26].

Endophytic microorganisms, defined as those found within plant tissue for at least part of their life cycle without causing disease under any known circumstances [27,28], have received a lot of attention in recent decades. Their ability to colonize an ecological niche similar to that of some phytopathogens enables them to act as potential biocontrol agents [9]. Among endophytic microorganisms, endophytic fungi are believed to have established relationships with plants approximately 400 million years ago [29]. However, approximately 95% of the fungal species that exist on the planet have not yet been described [30]. Endophytic fungi play a fundamental role in increasing plant tolerance to abiotic (e.g., greater tolerance to drought, metals, salt, and heat) as well as biotic factors by enhancing host resistance to insect feeding, deterring mammalian herbivore feeding, inducing systemic resistance, and conducting biocontrol through direct competition and/or the production of secondary metabolites with antibacterial and antifungal activity [31,32]. Various endophytic fungi (e.g., *Aspergillus clavatonanicus*,* Aureobasidium pullulans* and *Phoma terrestris*) and their metabolites (e.g., transoct-2-enal, N-amino-3-hydroxy-6-methoxyphthalimide and epoxycytochalasin H) isolated from different plant species (e.g., *Taxus mairei*,* Prunus avium* and *Panax ginseng*) have proven effective in inhibiting in vitro growth of *B. cinerea* [9]. *B. cinerea*‘s ability to develop endophytically before infecting the host plant was described [1]. This makes managing disease even more difficult [1]. The search for microorganisms that share the same lifestyle as *B. cinerea* is important. They constitute an interesting focus of research for the biocontrol of gray mold. It is possible that most of the fungi not yet described are found in the tropics considering that the location and environmental conditions where host plants develop have a major impact on the frequency and diversity of the population of endophytic fungi [30,33].

This study describes the process of isolating an endophytic fungus from *Gliricidia sepium*, a crop species in Colombia. It assesses the fungus’s biocontrol capabilities in vitro against *B. cinerea* and provides an analysis of the metabolites produced with this fungus during in vitro co-cultivation with *B. cinerea*.

## 2. Results

### 2.1. Isolation and Identification of Endophytic Fungi from Gliricidia sepium

Healthy leaf samples from *G. sepium* were collected from Barranquilla, Colombia (Figure 1). Several endophytic fungal morphotypes were isolated using an endophytic microorganism isolation protocol (described in Section 4.2), in which the outer surface of the plant material was sterilized. To assess the antifungal activity of the isolated endophytic microorganisms against *B. cinerea*, a confrontation in vitro assay (co-culture) was employed. Of these microorganisms, only one fungal isolate proved effective in inhibiting the mycelial growth of *B. cinerea*.

Subsequently, this endophytic fungus isolated from *G. sepium* was identified by the identification service of the Spanish Type Culture Collection (CECT), by means of both morphological and molecular methods, as *Sordaria tomento-alba*. The isolate exhibits the macroscopic and microscopic characteristics typical of the family Sordariacea especified species. Neighbor-joining phylogenetic analysis was conducted using the Kimura two-parameter model and a bootstrap test with 5000 runs (MegAlign, DNASTAR^®^ Lasergene package v. 7.1.0). The sequences of related fungal species/genus were downloaded from the GenBank database, from the family Sordariacea, to which *Sordaria* belongs. The phylogenetic trees shown in Figure 2 were constructed using (i) forty-six sequences, including six genera and twenty-five species, for the ribosomal DNA region comprising the intergenic spaces ITS1 and ITS2, including the 5.8S rRNA (Figure 2A); (ii) forty-three sequences, including five genera and twenty-four species, for the 28S rRNA gene (Figure 2B); and (iii) twenty-six sequences, including four genera and nineteen species, for the beta-tubulin gene (Figure 2C). Based on all these studies, it was determined that strain ST1-UCA is clearly grouped with the species *Sordaria tomento-alba* (Figure 2).

### 2.2. In Vitro Studies of Antagonistic Activity against Botrytis cinerea

#### 2.2.1. Antagonistic Activity Assay against *B. cinerea* during Co-Culture

*B. cinerea* B05.10 and *S. tomento-alba* ST1-UCA were placed together in an in vitro confrontation assay to evaluate the antagonistic effects of the endophytic fungus against the fungal pathogen. ST1-UCA exhibited a mean of 60% growth inhibition activity against *B. cinerea* during co-cultivation (Table 1 and Figure 3).

During co-cultivation, a dark-colored zone was observed in the back of the Petri dish in the area of interaction between the two microorganisms (Figure 3, red box). This suggests an increase in the production of metabolites in this area. This dark zone of interaction remained unchanged after several weeks of co-culture as neither of the microorganisms was able to overcome the other. This pattern coincides with that proposed by Bertrand et al. (2014), who noted that four main types of interactions can occur during a confrontation between microorganisms in a solid culture medium: (a) remote inhibition, (b) zone of confrontation (c) contact inhibition, and (d) over-growth [34]. A confrontation zone was clearly observed in the case of *S. tomento-alba* versus *B. cinerea* B05.10 (Figure 3, red box).

#### 2.2.2. Antifungal Effect of VOCs Produced with *S. tomento-alba* against *B. cinerea*

*B. cinerea* B05.10 and *S. tomento-alba* ST1-UCA were also co-cultured in the absence of physical contact to evaluate the antagonistic effects of the volatile organic compounds (VOCs) emitted by strain ST1-UCA, using the double disc method [35]. After incubation for 7 days at 25 °C, *B. cinerea* growth was significantly reduced compared to the control, demonstrating a clear inhibitory effect in the absence of direct contact between fungi. This result suggests that ST1-UCA produces VOCs with antifungal properties against this pathogen. As shown in Table 1 and Figure 4, the growth inhibition rate against *B. cinerea* exceeded 50%.

### 2.3. Metabolite Production during In Vitro Co-Culture

#### 2.3.1. Botryanes Production in Antagonist Test

*B. cinerea* B05.10 and ST1-UCA were inoculated on the same PDA plates (Figure 5a). The fractions obtained yielded botryanes weights of 1.01 ± 0.08 µg·mL^−1^ and 2.41 ± 0.53 µg·mL^−1^ for non-interaction and interaction zones, respectively, and 6.4 ± 2.14 µg·mL^−1^ in the control (Figure 5b). This is a statistically significant reduction (*p* < 0.05) in botryanes production in both the non-interaction and interaction zones. Additionally, changes in the hyphae of *B. cinerea* were observed in both zones, suggesting an impact on both areas (Figure 5c). Hyphae were found to be dematiaceous, non-septate, macrosiphonate, and sheet-like.

#### 2.3.2. Metabolites Produced with *S. tomento-alba* during Co-Cultivation with B05.10

As observed in Figure 6a, the zone where metabolites were extracted was identified as the confrontation zone. The crude extract (Figure 6b) shows that during its confrontation with *B. cinerea* in this zone, *S. tomento-alba* produced molecules that were apparently fungicidal or fungistatic. A spectroscopic study of the compounds isolated from the extract taken from the zone of confrontation with *B. cinerea* led to the identification of seven molecules (**1**–**7**) (Figure 7): trans-sordariol (**1**) [36,37,38], trans-sordarial (**2**) [36,39], 3′-episordariol (**3**), heptacyclosordariolone (**4**) [36], cyclosordariolone (**5**) [36], sordamentone A (**6**) and sordamentone B (**7**). Of these, heptacyclosordariolone (**4**) (Figure 7), a derivative of sordiarol and a typical metabolite of species of the genus *Sordaria* [36], was the most abundant molecule produced via ST1-UCA. Compounds **1**, **2**, **4**, and **5** were previously reported [36,37], and their absolute configurations were finally determined [38,39]. The ^1^H- and ^13^C-NMR data, and HMBC correlations appear in Appendix A.

The ^1^H and ^13^C NMR spectra of compound **3**, C_12_H_16_O_4_, had a pattern of signals that were characteristic and very similar to those of sordariol (**1**), except for the coupling constant of the signal corresponding to protons H-3′, δ 4.3 (1H, s(br)) and H-4′, δ 4.0 (1H, dd(br), *J =* 6.4, 3.6 Hz). This suggested that compound **3** was an epimer in the H-3′ of sordariol (**1**). The structure was confirmed with ^13^C NMR, COSY, HSQC, and HMBC experiments (Appendix A). The correlations observed in the HMBC spectrum were consistent with proposed structure **3**.

Compounds **6** and **7** exhibited very close and equally intense peaks in the HPLC chromatogram and shared the same molecular mass (*m*/*z* 221 [M − H]^+^, C_12_H_13_O_4_). ^13^C NMR spectra revealed the presence of a benzene ring, one methyl, two methylene groups, and two alkyl methines together with three quaternary carbons, one of which at δ 210.5 ppm, corresponded to a carbonyl group. The ^1^H NMR spectra were similar and revealed the presence of a 1,2,3-trisubstitued benzene ring, an oxygenated methylene group, and two spin systems corresponding to CH_3_-CH and CH_2_-CH fragments, both with methine carbons on an oxygenated function. The proposed structures were consistent with the correlations observed in the COSY, HSQC, and HMBC experiments. Both compounds could be separated via HPLC, where compound **6** was isolated pure, and compound **7** was obtained mixed with 50% of the other isomer. The difference between the two NMR spectra enabled us to assign the spectroscopic constants for each of these compounds (Appendix A). However, due to the small pure amount of compound **6** isolated, it was impossible to assign the stereochemistry at carbon C-1′. The configuration at the C-4′ carbon was assigned based on biogenetic considerations, with respect to the configuration of the other compounds isolated from this organism at this C-4′ carbon. Extensive COSY, HMBC, and HSQC experiments (Appendix A) prompted us to propose structures **6** and **7** for these compounds, which were named sordamentone A and B.

### 2.4. In Vitro Assessment of Antifungal Activity of Metabolites Produced with S. tomento-alba

#### 2.4.1. Inhibition of *B. cinerea* Mycelial Growth

The co-cultivation of two microorganisms typically leads to the isolation of new molecules. However, some studies have reported an increase in the production of a specific metabolite [40,41]. Serrano et al. (2017) proposed that co-cultures involving fungi and *B. cinerea* can activate cryptic pathways in fungi isolated from different plant environments and could lead to the characterization of new antifungal agents [42]. Figure 8 and Table 2 show that the compound heptacyclosordariolone (**4**) inhibited the mycelial growth of B05.10 starting at a concentration of 5 μg·mL^−1^ (37%) and reached 100% inhibition at a concentration of 250 μg·mL^−1^. The IC_50_ was calculated from the data obtained in each of the concentrations tested and yielded a value of 11.65 μg·mL^−1^. These results demonstrate that heptacyclosordariolone (**4**) is an effective in vitro development inhibitor against *B. cinerea* mycelium.

#### 2.4.2. Microtiter Plate *B. cinerea* Spore Germination Assay

Based on the activity shown by heptacyclosordariolone (**4**) in the MGI assays, its ability to inhibit the germination of *B. cinerea* conidia was evaluated. Table 3 shows that after 48 h of incubation, heptacyclosordariolone (**4**) inhibited spore germination starting at a concentration of 5 μg·mL^−1^, reaching 100% inhibition at 140 μg·mL^−1^. Therefore, 5 μg·mL^−1^ was the minimum concentration of heptacyclosordariolone needed to affect the normal spore germination process under the assay conditions (Figure 9a).

To determine whether the germination inhibition effect was maintained for more than 48 h, the samples were observed under an inverted microscope for 120 h and the same behavior was observed, indicating that the compound remained active for that period of time (Figure 9b).

As shown in Figure 9a, the spores exposed to heptacyclosordariolone (**4**) exhibited granulations and structural changes with a cytoplasmic retraction, although some appeared to have a normal structure. When the conidia exposed to the compound were subsequently inoculated in a PDA medium (without the compound), they remained viable and developed mycelium normally after 120 h of incubation, as shown in Figure 9c. This indicates that this compound, at the tested concentrations, acts as a fungistatic agent preventing the germination of conidia without affecting their viability.

## 3. Discussion

### 3.1. Endophytic Fungi an Essential Part of the Plant Microbiome

More than one million endophytic fungi are ubiquitous and live within host plants without causing any noticeable symptoms of disease, being found in all plant species (≈300,000) at a ratio of 1:4 or 1:5 fungi per host [43,44]. These fungi are considered a rich source of molecules with antimicrobial activities and they are able to help their host plants adapt to both biotic and abiotic stress conditions [32,44]. Several endophytic fungi isolated from different plant species have been evaluated as possible biocontrol agents of *B. cinerea*, and different molecules have been isolated that have exhibited in vitro activity against gray mold [9]. However, to date, no endophyte fungus isolated from the plant *G. sepium* has been evaluated as a possible biocontrol agent of *B. cinerea*. Furthermore, *G. sepium* is a leguminous tree belonging to the *Fabaceae* family and is native to Central America and northern South America. It is currently distributed throughout tropical America, the Caribbean, Africa, Asia, and the Pacific islands, in areas between zero and 1300 m above sea level, with rainfall between 600 and 6000 mm·year^−1^ [45]. *G. sepium* leaves are used as a repellent against ectoparasites and to treat skin diseases, while the seeds and bark have rodenticidal qualities. In Colombia, the plant is commonly known as “*matarratón*” (rat poison) [45]. *S. tomento-alba*, the endophytic fungus isolated from *G. sepium*, taxonomically belongs to the class *Sordariomycetes*, the second largest class of the phylum *Ascomycota*, which includes 6 subclasses, 32 orders, 1331 families, and more than 10,000 species [46]. The *Sordariomycetes* class includes important plant pathogens but also saprophytes, endophytes, epiphytes, and mycoparasites, among others [47]. Some species are used as biocontrol agents (such as *Beauveria bassiana* and *Trichoderma viride*, among others) in the production of bioactive compounds with clinical applications and in the biotechnology industry [47]. Several species of *Sordaria* have been isolated from the endospheres of different plant species, as reported by Yan et al. [48], such as kiwi plants (González and Tello (2011)), *Vitis vinifera* [49] (De Errasti et al. (2010)), and *Ligustrum lucidum* [50]. However, there are no reports of *S. tomento-alba* isolated as an endophyte of *G. sepium*, nor has its effectiveness as a biocontrol agent against *B. cinerea* been evaluated. Hence, this communication is a first-time description of a new biological control agent against gray mold.

### 3.2. The Co-Culture of B. cinerea vs. S. tomento-alba

A confrontation zone was observed during the co-cultivation of *B. cinerea* and ST1-UCA (Figure 3). This dark area was located in the zone where the two microorganisms interact (interaction zone), as described by Bertrand et al. (2014) [34]. This suggests that the two microorganisms are engaged in “chemical warfare” through the production of secondary metabolites. These metabolites may act as transcriptional regulators or modifiers of the biosynthetic pathways of competing microorganisms [34]. In this regard, the effect of the botrydial phytotoxin is not limited to plants; it has also exhibited antimicrobial and cytotoxic activity, and increased production has been observed during co-cultivation with biocontrol microorganisms [13,14,51]. Test results show that botryanes production with *B. cinerea* increases in the interaction zone in the co-culture with ST1-UCA. This could indicate that the phytopathogen takes advantage of the antimicrobial activity of botrydial to defeat or control the growth of *S. tomento-alba*, which would explain why ST1-UCA stops growing in this area. This phenomenon has already been described in studies performed by Malmierca et al. (2014), where they analyzed the confrontations between *Trichoderma arundinaceum* and *B. cinerea* within in vitro cultures. These authors showed how fungi established a bidirectional transcriptional regulation between them [51]. It is therefore likely that some of the metabolites produced via ST1-UCA could repress the production of botryanes and, at the same time, the botryanes could repress some biosynthetic pathways in ST1-UCA. Test results also revealed a low production of botryanes via *B. cinerea* along non-confrontation zones (Figure 3). This may be because *B. cinerea* does not directly interact with *S. tomento-alba* in this area. Therefore, the cells of this region of the mycelium are not forced to produce botryanes as there is no direct interaction with ST1-UCA. These results are different from those obtained by Bolivar et al. (2020) [14] and Vignatti et al. (2020) [13], who observed an increase in botryanes in this zone during co-cultivation with *Bacillus subtilis* and *Bacillus amyloliquefaciens*, respectively, even in the unconfronted zone. It is also possible that the low production in this case could be due to the fact that the margin of mycelial growth in this area reaches the edge of the Petri dish and nutrient depletion could have an effect on botrydial synthesis, taking into account the results of Liñeiro et al. (2018), which demonstrated that the presence of glucose in the medium promotes the production of botrydial [52]. Our double disk test results show that ST1-UCA is capable of producing VOCs and, since these compounds can extend their activity both at close distances by aqueous diffusion and to more distant areas via airborne diffusion, these compounds could be responsible for the repression of botryanes synthesis in the unconfronted zone [53]. VOCs could also be related to the changes in the morphology of the hyphae observed in this area of the Petri dish. In this regard, the studies conducted by Li et al. (2012) demonstrated that the volatile compounds produced via *Streptomyces globisporus* JK-1 are capable of crossing the fungal wall and altering the permeability of the plasma membrane in *B. cinerea* [54]. In addition, the studies by Soylu et al. (2010) showed that the volatile fraction of essential oils extracted from plants from the *Lamiacea* family were capable of altering the hyphae of *B. cinerea*, which presented coagulation and cytoplasmic vacuolization, among other morphological changes [55].

### 3.3. Metabolites’ Production during Co-Culture

Given that mycelial growth is faster in *S. tomento-alba* than *B. cinerea*, and following the results of Martín et al. (2015) who isolated an endophytic strain of *Sordaria* sp. from *Ulmus minor* whose inhibition capacity against *Ophiostoma novo-ulmi* was due to substrate competition, this same mechanism could be responsible for *S. tomento-alba*’s inhibition against *B. cinerea* [56]. However, the presence of a confrontation zone characterized by dark coloration like that observed in the co-culture between *B. cinerea* and ST1-UCA (Figure 3) suggests that, in addition to stunting *B. cinerea* by preventing the use of nutrients needed for growth, ST1-UCA is able to produce metabolites with antimicrobial activity [34]. In this connection, the naturally occurring chemoecological relationships between microorganisms can be exploited in vitro through co-culture experiments (confrontation) to discover new metabolites and gain insights into the triggering of specific biosynthetic pathways, which are mainly related to defense mechanisms [34]. In this sense, Bouillant et al. (1989) studied the secondary metabolism of the fungi species *Sordaria macrospora* [36]. The research focused on isolating and characterizing the compound heptacyclosordariolone (4). Tsague et al. (2020) isolated the compounds identified as heptacyclosordarianone, heptacyclosordariolone, and sordariol from a strain of *Sordaria* sp. AM-71, an endophyte of *Garcinia polyantha* [57]. Heptacyclosordariolone showed antibacterial capacity against the strains of *Bacillus subtilis*, *Pseudomonas agarici*, and *Micrococcus luteus* [57]. However, its antifungal activity against phytopathogens was not studied [36,57]. Furthermore, Li et al. (2016) isolated the endophytic fungi *Sordaria macrospora* from the bark of the plant *Ilex cornuta*. These fungi produced sordariol and sordariol derivatives: 12-methoxy sordariol, bisordariols A-D, xyralinol A, and two isobenzofuranyl derivatives. This study established the strong antioxidant capacity of these compounds, especially the bisordariol derivatives [38].

Sordarin, an antifungal diterpene that inhibits protein synthesis in fungi, was isolated from *Sordaria arenosa*. This compound acts on elongation factor 2, which is in charge of promoting the translocation of the ribosome on the mRNA. This translocation is inhibited by sordarin, which stabilizes the ribosome-elongation factor 2 complex, thereby preventing peptide chain formation [58]. This shows that the genus *Sordaria* is a source of bioactives molecules with potential beneficial applications in medicine and agriculture, as recently reported by Charria-Girón et al. [59].

In this article, from the strain ST1-UCA, seven metabolites were isolated for the first time—the known compounds comprising trans-sordariol (**1**), trans-sordarial (**2**), heptacyclosordariolone (**4**), and cyclosordariolone (**5**) [36,37,38,39,57], and the new metabolites consisting of 3′-episordariol (**3**), sordamentona A (**6**), and sordamentona B (**7**), (Figure 7)—which were produced via *S. tomento-alba* in co-culture with *B. cinerea*. Heptacyclosordariolone (**4**) was produced in a greater quantity than the other metabolites which were minor. Compound **4** was successful in inhibiting both mycelial growth and the germination of *B. cinerea* conidia, but its action mechanisms are still unknown. In fact, it has been reported that 51% of the molecules isolated from endophytic fungi were unknown [60].

The United Nations Food and Agricultural Organization (FAO) estimates that one-third of the food produced in the world is lost in the post-harvest period, with the main threat being disease caused by pathogens [61]. Specifically, *B. cinerea* is one of the main causes of fresh fruit and vegetable losses, accounting for 40% of food loss worldwide, pointing to the importance of seeking strategies to control the post-harvest damage caused by this fungus [61,62]. Based on the results obtained here with heptacyclosordariolone (**4**), its use as a natural antifungal agent for post-harvest crops could be proposed.

## 4. Materials and Methods

### 4.1. Strains and Culture Conditions

Two strains were used in this study: the wild-type *B. cinerea* B05.10 and one strain of fungus identified in this study as *S. tomento-alba* (Table 4). Potato dextrose agar (PDA) medium (Condalab S.A., Torrejón de Ardoz, Madrid, Spain) was used for routine fungal cultures. The cultures were incubated at 25 °C under 24 h daylight.

### 4.2. Isolation and Identification of Endophytic Fungi from Gliricidia sepium

Healthy leaves from *G. sepium* were sampled from Barranquilla, Colombia (Colombia’s Atlántico Department). Fresh samples were brought to the laboratory in sterile packaging and immediately processed. Leaves were washed with sterile distilled water (SDW), sterilized using 80% ethanol for 1 min, and then washed with 4% NaOCl for 5 min. Leaves were then washed eight successive times using SDW [64]. Several aliquots of the final rinse water were grown on PDA agar plates to confirm the absence of any microbial growth and therefore in a correct surface sterilization condition. Sterilized leaves were macerated with 3 mL of sterile 0.9% NaCl in a sterile mortar. The macerate dilution and tissue segments were placed on PDA agar medium and incubated at 25 °C for 5 days [64]. Fungal colonies were selected and streaked on PDA plates under axenic cultures. The isolates were maintained on PDA agar at 25 °C for routine experiments and spores were stored in 60% (*v*/*v*) glycerol at −20 °C for later studies.

The endophytic fungus isolated from *G. sepium* was identified as *Sordaria tomento-alba* using the service of the Spanish Type Culture Collection (CECT, https://www.uv.es/cect (accessed on 10 November 2020), based on both phenotypic and molecular techniques. Three regions of the fungal genome were amplified with conventional PCR: (a) amplification and sequencing (with readings in both directions) of the ribosomal DNA region comprising the intergenic spaces ITS1 and ITS2, including the 5.8S rRNA (ITS5 5′-GGAAGTAAAAGTCGTAACAAGG-3′; ITS4 (5′-TCCTCCGCTTATTGATATGC-3′); (b) partial amplification and sequencing (with readings in both directions) of the 28S rRNA gene (LR0R 5′-GTACCCGCTGAACTTAAGC-3′; LR7 5′-TACTACCACCAAGATCT-3′); and (c) partial amplification and sequencing of the β-Tubulin gene (with readings in both directions) (Bt2a 5′-GGTAACCAAATCGGTGCTGCTTTC-3′; Bt2b 5′-ACCCTCAGTGTAGTGACCCTTGGC-3′). The sequencing of these regions was compared with those in NCBI databases. Sequences were submitted to NCBI database with the next accession number: OR835245 for ITS region; OR835246 for *28S rRNA* gene; and OR879043 for *β-Tubulin* gene. To study the phylogenetic relationship of our isolate, more sequences of related genera and species were downloaded from the GenBank database and included in the phylogenetic trees.

### 4.3. In Vitro Studies of Antagonistic Activity against Botrytis cinerea

#### 4.3.1. Antagonistic Activity Assay against *Botrytis cinerea* during Co-Culture

To evaluate antagonistic effects against *B. cinerea* B05.10, fungi isolated from *G. sepium* were assayed on 150 mm PDA plates. Five-millimeter mycelial discs of seven-day-old cultures of each fungus were inoculated on opposite sides of the Petri dish. All conditions were assayed in triplicate. Antagonistic assays were incubated at 25 °C for 6 days under 24 h of daylight.
$$\%\;antagonistic\;activity:\left(1-\frac{Rg}{Rc}\right)\times 100$$

The antagonistic effect was calculated as described by Kusari et al. (2013) [65]. *Rc* represents the mean value of *B. cinerea* radial growth in the absence of the antagonist, while *Rg* represents the mean value of *Botrytis cinerea* radial growth during the antagonistic test.

#### 4.3.2. Antifungal Effect of VOCs Produced via *S. tomento-alba* against *B. cinerea*

The antifungal activity of volatile organic compounds (VOCs) emitted by strain ST1-UCA was assessed against *B. cinerea* B0510 by means of a triplicate two-sealed base plate assay. To that end, a five-day-old mycelial plug (7 mm) from *B. cinerea* B05.10 was inoculated on one base plate containing PDA media and another similar mycelium plug from ST1-UCA was placed on another base plate containing PDA media. One plate was inoculated with pathogenic fungi and the other served as a control with sterile water. These two base plates were sealed with a double layer of parafilm and cultured at 25 °C for 7 days under 24 h of daylight. The diameters of the *B. cinerea* colonies were measured when the fungal hyphae reached the edge of the control plates. The mycelial growth inhibition percentage was calculated based on the difference between the treatment and control groups [35].

### 4.4. Metabolites Production during Co-Culture

#### 4.4.1. Botryanes Production in Antagonist Test

Botryanes production (botrydial + dihydrobotrydial) was studied during the co-culture of the B05.10 strain and antagonistic ST1-UCA. This study was conducted in triplicate and on PDA plates where B05.10 and antagonistic fungi were inoculated at opposite sides of the plates and incubated at 25 °C for 7 days under 24 h of daylight. Twelve PDA plugs containing mycelia (1 cm diameter plugs) were then taken from two different sites: (i) from the fungus–fungus interaction zone and (ii) from the non-interaction part of the B05.10 furthest from ST1-UCA.

Botryanes were extracted following the protocol optimized by Izquierdo-Bueno et al. (2018) [66]. Briefly, 12 PDA plugs were extracted with ethyl acetate (3 × 300 mL) by means of an ultrasonic bath for 30 min. The ethyl acetate organic extract was dried over Na_2_SO_4_, concentrated to dryness, and then separated in a chromatography column (silica gel) eluted with ethyl acetate–hexane (40:60). The isolated botryanes were identified with thin-layer chromatography and characterized via ^1^H-NMR. The production of botryanes was expressed in micrograms of botryanes per millilitre of medium (µg botryanes.mL^−1^).

#### 4.4.2. Metabolites Produced via *S. tomento-alba* during Co-Cultivation with *B. cinerea*

The antagonist assays were carried out in triplicate, and a single culture of B05.10 fungus was incubated under the same conditions for comparative purposes and as a control. Experimental conditions were as described in Section 4.4. PDA plates (150 mm) were inoculated with plugs (7 mm) containing five-day-old mycelia from *B. cinerea* B05.10 and ST1-UCA, positioned on opposite sides of the plates and were then incubated at 25 °C for 7 days under 24 h of daylight conditions. Mycelium plugs from interaction and non-interaction zones of *B. cinerea* were then collected, extracted with ethyl acetate and evaporated to dryness in a rotary evaporator following the protocol optimized by Izquierdo-Bueno [66]. The crude extract was separated using column chromatography (Merck silica grain 60–200 microns) and using a mixture containing increasing percentages of hexane/ethyl acetate (10–100%) as the eluent. Semi-preparative and analytical HPLC purification was performed on a Merck-Hitachi LaChrom liquid chromatograph equipped with an L-7490 refractive index and an L-7100 pump, using Primayde System software (version 1.0). The eluents used were hexane and ethyl acetate, as well as mixtures of these, which were previously degassed and filtered through 0.45 μm pore size Millipore filters. Compound structures were elucidated by analyzing the 1D (1H-, ^13^C-NMR) and 2D (COSY, HSQC, HMBC) NMR spectra performed on Agilent-400 MHz and Agilent-500 MHz equipment at a temperature of 25 °C. Chemical shifts were expressed on the δ scale in ppm and coupling constants (*J*) in hertz (Hz). The δ values were referenced to the residual peak of chloroform and methanol for proton at δH 7.25 and δH 5.30 ppm, respectively, and for carbon at δC 77.0 and δC 49.0 ppm, respectively [51].

The known metabolites sordariol (**1**) [36,37,38], sordarial (**2**) [37,39], heptacyclosordariolone (**4**) [36], and cyclosordariolone (**5**) [36] were isolated and characterized via comparing spectroscopic data with those described in the literature (Appendix A). Additionally, three new metabolites—3′-episordariol (**3**), sordamentone A (**6**) and sordamentone B (**7**)—were isolated and characterized.

#### 4.4.3. Spectroscopic Data for Compounds **3**, **6**, and **7**

**3′-Episordariol (3)**: yellow oil, IR (film) *ν*_max_ 3338, 2974, 2921, 1582, 1468, 1373, 1275, 1075, 977, 789, 754 cm^−1^; ^1^H NMR data (CDCl_3_, 500 MHz) δ 7.16 (t, 1H, *J* = 7.8 Hz, H-4), 6.97 (dd, 1H, *J* = 7.8, 1.1 Hz, H-5), 6.85 (d, 1H, *J* = 15.7 Hz, H-1′), 6.83 (d(superimposed), 1H, *J* = 7.9 Hz, H-3), 6.1 (dd, 1H, *J* = 15.7, 6.6 Hz, H-2′), 5.0 (s, 2H, H-7), 4.3 (s (br), 1H, H-3′), 4.0 (dd(br), 1H, *J* = 6.4, 3.6 Hz, H-4′)), 1.20 (d, 3H, *J* = 6.5 Hz, H-5′). ^13^C NMR (CDCl_3_, 125 MHz) δ 156.5 (C, C-2), 136.2 (C, C-6), 131.0 (CH, C-2′), 129.6 (CH, C-1′), 129.1 (CH, C-4), 121.8 (C- C-1), 118.9 (CH, C-5), 116.2 (CH, C-3), 76.2 (CH, C-3′), 70.3 (CH, C-4), 60.2 (CH_2_, C-7), 17.7(CH_3_, C-5′). HREIMS *m/z* 223.0983 [M − H]^+^ (calcd for C_12_H_15_O_4_, 223.0970).**Sordamentone A (6)**: oil, IR (film) *ν*_max_ 3337, 2863.5, 1716, 1628, 1600, 1470, 1371, 1294, 1004, 826, 781, 712 cm^−1^; ^1^H NMR (CDCl_3_, 500 MHz) δ 7.16 (dd, 1H, *J* = 8.1, 7.4 Hz, H-4), 6.73 (dt, 1H, *J* = 7.4, 0.7 Hz, H-5), 6.68 (dt, 1H, *J* = 8.1, 0.7 Hz, H-3), 5.72 (q, 1H, J = 3.6, 2.8 Hz, H-1′), 5.15 (dd, 1H, *J* = 12.3, 2.8 Hz, H-7a), 5.07 (dt, 1H, *J* = 12.3, 1.8 Hz, H-7b), 4.27 (q, 1H, *J* = 7.2 Hz, H-4′), 3.06 (dd, 1H, *J* = 16.0, 7.8 Hz, H-2′a), 2.86 (dd, 1H, *J* = 16.0, 4.7 Hz, H-2′b), 1.41 (d, 3H, *J* = 7.2 Hz, H-5′). ^13^C NMR (CDCl3, 125 MHz) δ 210.5 (C, C-3′), 150.1 (C, C-2), 143.1 (C, C-6), 129.5 (CH, C-4), 125.3 (C, C-1), 114.4 (CH, C-3), 113.4 (CH, C-5), 80.5 (CH, C-1′), 73.4 (CH, C-4′), 70.8 (CH_2_, C-7), 44.5 (CH_2_, C-2′), 19.3 (CH_3_, C-5′). HREIMS *m/z* 221.0822 [M − H]^+^ (calcd for C_12_H_13_O_4_, 221.0814).**Sordamentone B (7)**: oil, IR (film) *ν*_max_ 3337, 2863.5, 1716, 1628, 1600, 1470, 1371, 1294, 1004, 826, 781, 712 cm^−1^; ^1^H NMR (CDCl_3_, 500 MHz) δ 7.167 (dd, 1H, *J* = 8.0, 7.5 Hz, H-4), 6.75 (d(br), 1H, *J* = 7.5 Hz, H-5), 6.68 (d(br), 1H, *J* = 8.0 Hz, H-3), 5.72 (m, 1H, H-1′), 5.15 (dd, 1H, *J* = 12.3, 2.6, 1.1 Hz, H-7a), 5.07 (dt, 1H, *J* = 12.3, 2.6 Hz, H-7b), 4.30 (q, 1H, *J* = 7.1 Hz, H-4′), 2.96 (dd, 1H, *J* = 16.0, 8.0 Hz, H-2′a), 2.91 (dd, 1H, *J* = 16.0, 4.4 Hz, H-2′b), 1.38 (d, 3H, *J* = 7.1 Hz, H-5′). ^13^C NMR (CDCl_3_, 125 MHz) δ 210.5 (C, C-3′), 150.1 (C, C-2), 143.1 (C, C-6), 129.5 (CH, C-4), 125.3 (C, C-1), 114.4 (CH, C-3), 113.4 (CH, C-5), 80.5 (CH, C-1′), 73.4 (CH, C-4′), 70.8 (CH_2_, C-7), 44.5 (CH_2_, C-2′), 19.3 (CH_3_, C-5′). HREIMS *m/z* 221.0825 [M − H]^+^ (calcd for C_12_H_13_O_4_, 221.0814).

### 4.5. In Vitro Assessment of Antifungal Activity of Metabolites Produced with S. tomento-alba

#### 4.5.1. Inhibition of *B. cinerea* Mycelial Growth

To determine the mycelial growth inhibition (MGI) by metabolites isolated from ST1-UCA (see Section 2.4.1), plates containing 10 mL of PDA medium supplemented with increasing amounts of the inhibitory substances (diluted in ethanol) under study were prepared. Negative controls were prepared using an equal quantity of the ethanol that was used to dilute inhibitory substances. Positive controls were prepared using dicloran at different concentrations (0,5-1-2,5-5-15-35-70 μg·mL^−1^). All PDA plates were inoculated using five-day-old mycelial plugs (7 mm) from *B. cinerea* B05.10. The plates were incubated at 25 °C for 7 days under 24 h of daylight, and radial mycelial growth was measured daily for 4 days. All conditions were assayed in triplicate.
$$MGI:\frac{dc-dt}{dc}\times 100$$

IMG was calculated as described by Simionato et al. (2017) [67]. In the expression, dc (mm) represents the mean value of *B. cinerea* radial growth in negative control plates, and dt (mm) represents the mean value of *B. cinerea* radial growth in each treatment. The 50% effective dose (ED50) was determined through regression analysis when growth was reduced by 50% compared to the control.

#### 4.5.2. Microtiter Plate *B. cinerea* Spore Germination Assay

To evaluate the effects of metabolites isolated from ST1-UCA on *B. cinerea* germination, increasing concentrations of organic compounds were dispensed into 96-well polystyrene microtiter plates.

*B. cinerea* B05.10 was grown at 25 °C under 24 h of daylight on PDA Petri dishes for 15 days. Subsequently, 20 mL of SDW was added to each plate and the surface was gently scraped with a sterile loop to release the spores. The resulting spore suspension was filtered using 30 μm filters (Nytal; Maissa, Barcelona, Spain). Conidia were pelleted via centrifugation at 5000× *g* for 5 min and then resuspended in SDW to achieve a final concentration of 1 × 10^6^ conidia mL^−1^. Aliquots of the conidial suspension were added to each well in the microtiter plates and mixed at a 1:1 dilution with the inhibitory substances (diluted in ethanol) being studied. Negative control well were included in each plate consisting of wells with 10% ethanol but no inhibitory substances, as were control wells void of both inhibitory substances and spores. All conditions were assayed in triplicate.

The microtiter plates were incubated at 25 °C, and absorbance measurements were taken at 570 nm and 600 nm during a 48 h period. The minimum inhibitory concentration was determined as the lowest concentration that did not exhibit turbidity after 48 h of incubation at 25 °C. Turbidity was interpreted as visible growth of B05.10. Subsequently, all the wells were plated on PDA agar to assess the viability of the spores [68].

### 4.6. Statistical Analysis

Experiments were conducted in triplicate (n = 3). Data were analyzed with Origin 9.3 software. A one-way analysis of variance and Tukey’s HSD multiple comparison test were employed to determine significance levels with a probability of *p* < 0.05.

## 5. Conclusions

In recent decades, agricultural systems have been striving to become more sustainable and reduce their environmental impact. Consequently, there has been a growing interest in the search for microorganisms with biocontrol capabilities. Among these microorganisms, endophytes have garnered significant attention from researchers due to their capacity to enhance plant growth through both direct and indirect mechanisms. It is believed that the close symbiotic relationship between endophytes and their host plants enables them to produce a wide range of molecules with diverse biological activities.

In this study, we successfully isolated the endophytic fungus *Sordaria tomento-alba* (ST1-UCA) for the first time from *Gliricidia sepium* plants cultivated in Colombia. Moreover, this study is the first to reveal the potential of the endophytic fungus *S.tomento-alba* as a promising biocontrol agent against the important phytopathogen *B. cinerea*. Our analyses show that this endophyte produces metabolites such as heptacyclosordariolone (**4**) when interacting with *B. cinerea* in co-culture. This provides novel molecular evidence of the triggered chemical pathways involved in the antagonistic activity of *S. tomento-alba*. Furthermore, ST1-UCA was successful in inhibiting the in vitro growth of *B. cinerea* and reducing the production of botryanes. Although no data are shown regarding the efficacy of this endophyte and its metabolites in suppression of gray mold disease, the ability of heptacyclosordariolone to inhibit the mycelial growth and conidia germination of *B. cinerea* was fully confirmed through in vitro bioassays. These experiments shed light on the potential fungistatic mechanisms of action of this compound against this phytopathogen. Further genomic and metabolomic studies could shed more light on the biosynthesis of these bioactive compounds in *S. tomento-alba* in response to pathogenic stimuli.

Although Colombia is one of the 14 countries with the highest biodiversity index in the world [69], few studies have explored microorganisms associated with plants (including rhizospheric, endophytic, and epiphytic microorganisms). By integrating modern chemical, genomic, and biological techniques, this work significantly advances our understanding of the complex interactions occurring between endophytes and phytopathogens at a molecular level. It also highlights the yet untapped potential of such chemical–biological relationships to discover new sustainable alternatives to replace synthetic fungicides in agriculture. This opens new avenues for research, offering opportunities to not only unveil the microbial diversity within this country but also to discover microorganisms with potential applications in biocontrol, plant growth promotion, enzyme production, bioactive compound synthesis, and more. Overall, this study makes key contributions to realizing the potential of endophyte-based biocontrol, moving towards more environmentally friendly and natural agricultural practices globally.

## Figures and Tables

**Figure 1 ijms-25-01022-f001:**
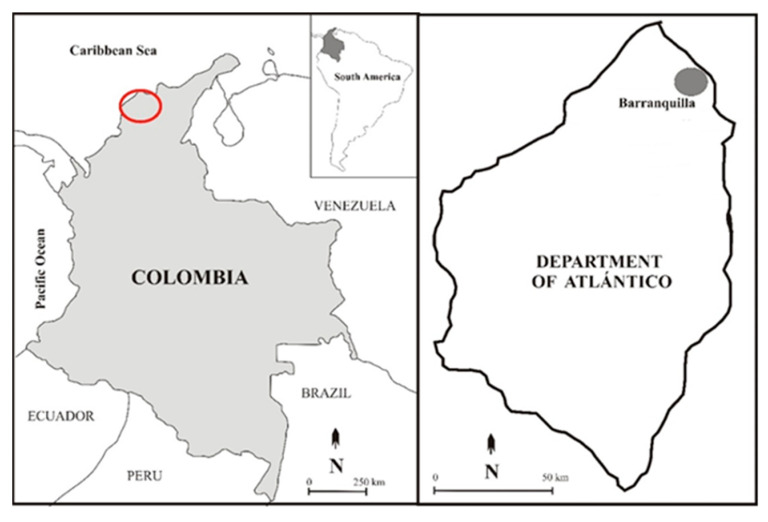
Atlántico Department of Colombia. On the left: a map of Colombia with a red mark showing the region from which the sample was taken. On the right: Colombia’s Atlántico Department with the city of Barranquilla.

**Figure 2 ijms-25-01022-f002:**
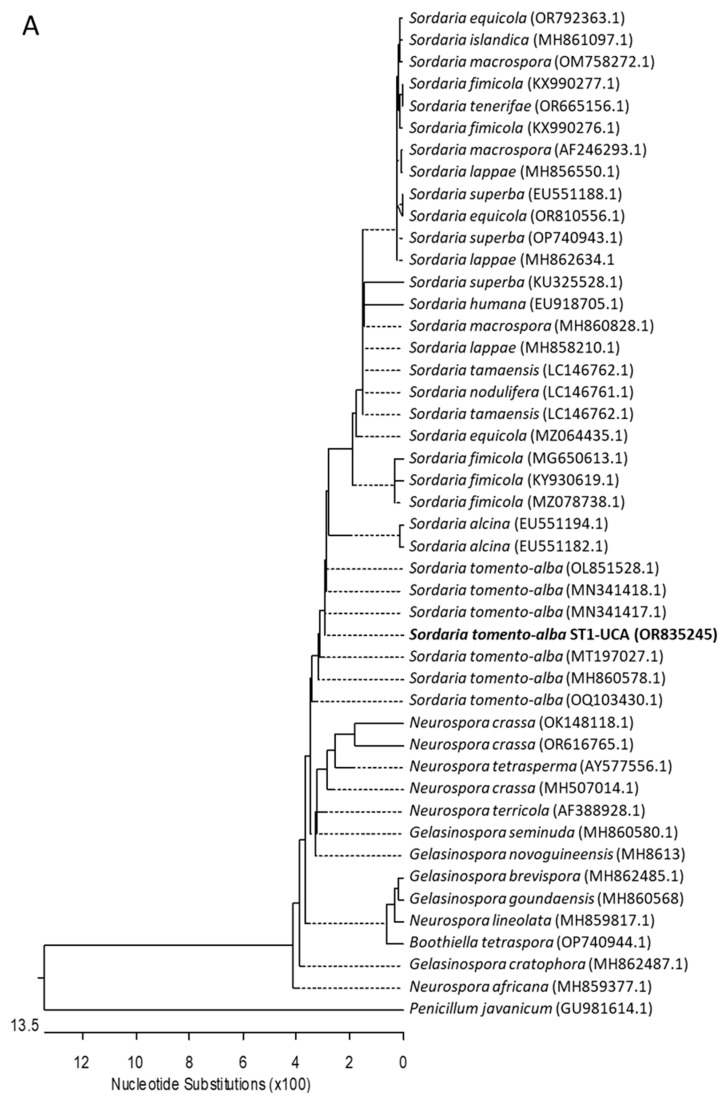
Neighbor-joining trees constructed using (**A**) intergenic spaces ITS1 and ITS2, including the 5.8S rRNA; (**B**) 28S rRNA gene; and (**C**) beta-tubulin gene sequences. Sequences identified in this study are highlighted in bold, and published sequences were obtained from the GenBank database. The length of each branch pair reflects the distance between respective sequence pairs. A dotted line on the tree denotes a negative branch length, while the bar indicates the number of nucleotide substitutions.

**Figure 3 ijms-25-01022-f003:**
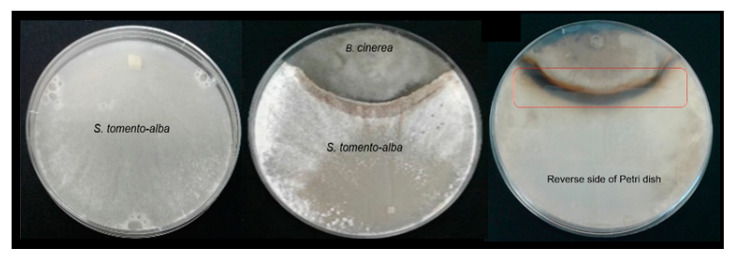
*S. tomento-alba* ST1-UCA strains in axenic culture (on the left); *S. tomento-alba* ST1-UCA vs. *B. cinerea* (on the right).

**Figure 4 ijms-25-01022-f004:**
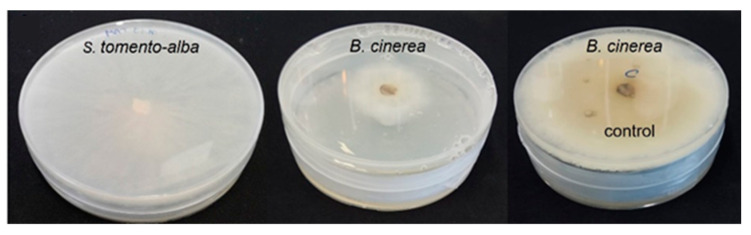
*S. tomento-alba* ST1-UCA strain in axenic culture (on the left); *S. tomento-alba* ST1-UCA vs. *B. cinerea* (in the middle); *B. cinerea* strain in axenic culture (on the right).

**Figure 5 ijms-25-01022-f005:**
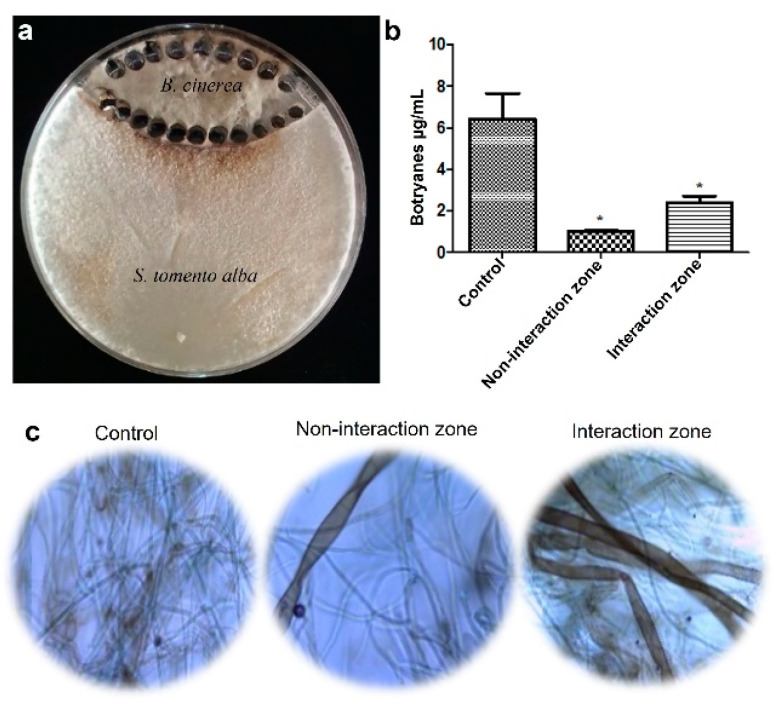
Co-culture of *S. tomento-alba* and *B. cinerea* B05.10. (**a**) Parts described as “non-interaction and interaction zones”; (**b**) production of botryanes from non-interaction and interaction zones in comparison to the *B. cinerea* B05.10 control. Data are expressed as the mean of three replicates ± SDs. Asterisks indicate statistically significant differences of each treatment when compared to the control (*p* < 0.05). (**c**) Microscopic characteristics of B05.10 hyphae from control, non-interaction, and interaction zones.

**Figure 6 ijms-25-01022-f006:**
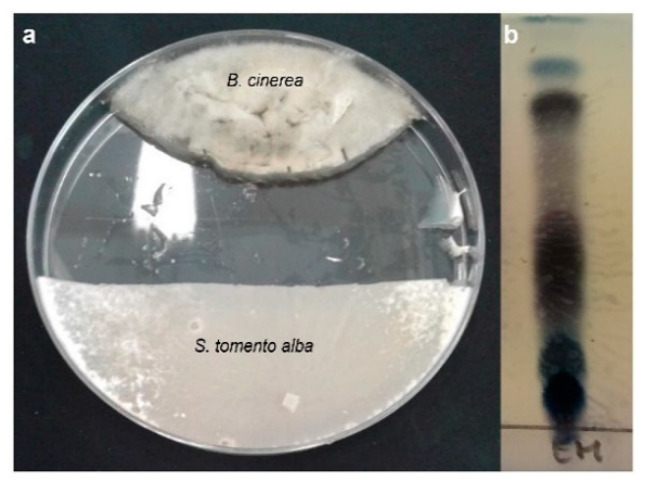
(**a**) Extraction zone of metabolites from *S. tomento-alba*; (**b**) TLC of the extract from *S. tomento-alba*.

**Figure 7 ijms-25-01022-f007:**
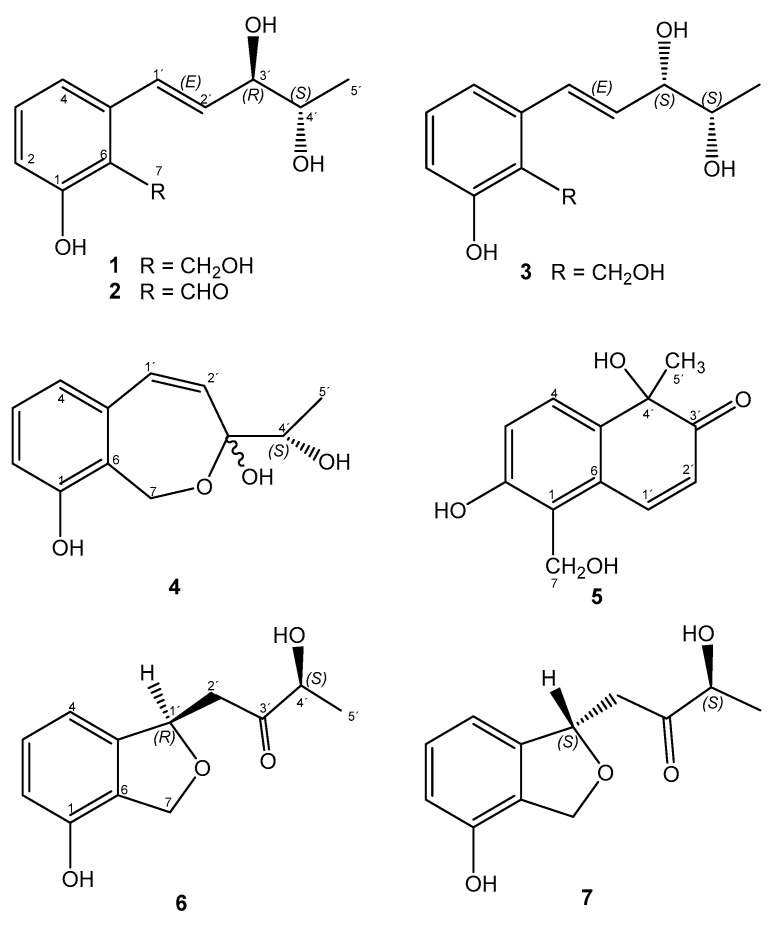
Metabolites isolated from the confrontation zone of *Sordaria tomento-alba* in co-culture with *B. cinerea*.

**Figure 8 ijms-25-01022-f008:**
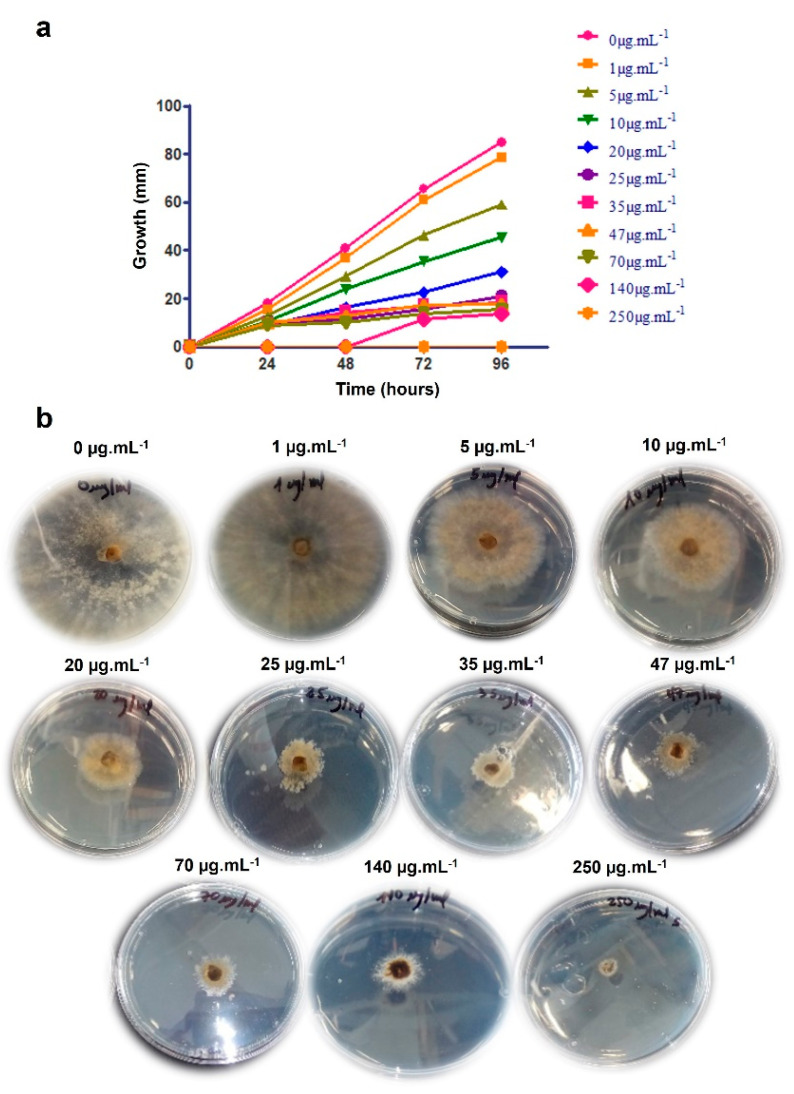
(**a**) *B. cinerea* mycelial growth during incubation with different concentrations of heptacyclosordariolone (**4**); (**b**) MGI assay after 96 h incubation at 25 °C.

**Figure 9 ijms-25-01022-f009:**
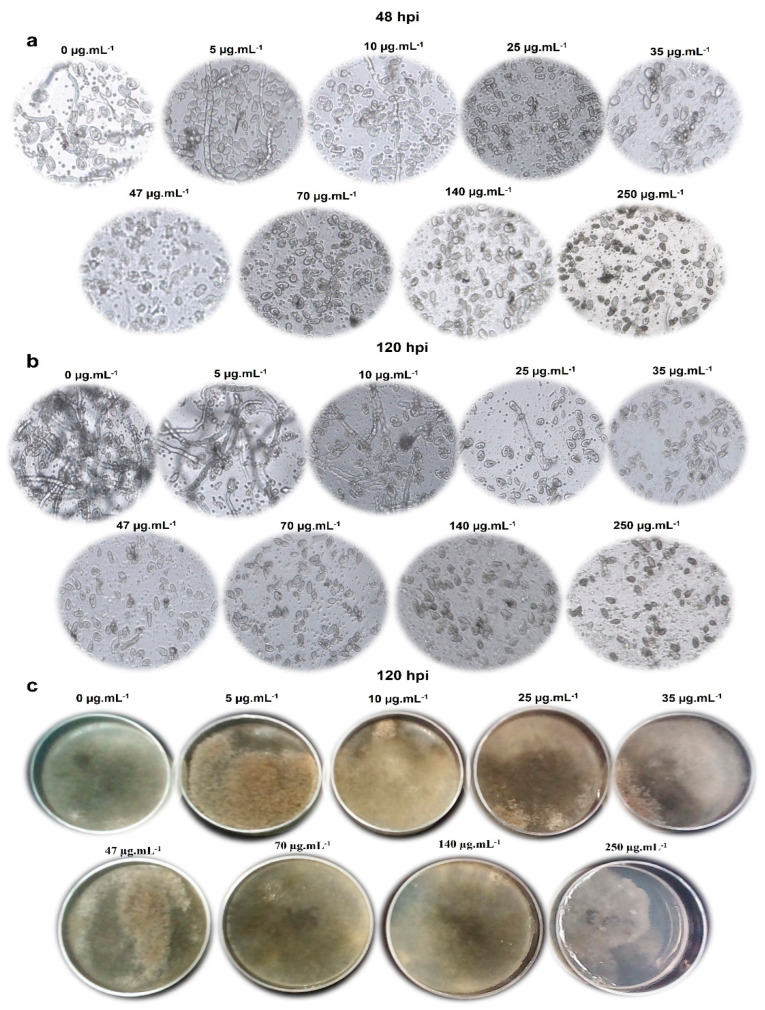
(**a**) Spores 48 hpi with heptacyclosordariolone (**4**); (**b**) spores 120 hpi with heptacyclosordariolone (**4**); (**c**) subcultures of spores (120 hpi) in PDA.

**Table 1 ijms-25-01022-t001:** Antagonistic effects against *B. cinerea* B05.10.

**Antagonism Test**	**% Inhibition**
Co-culture	66
Double disc co-culture	53

**Table 2 ijms-25-01022-t002:** Percentage of mycelial growth inhibition via heptacyclosordariolone.

Mycelial Growth Inhibition [MGI]										
[ ] μg·mL^−1^	1	5	10	20	25	35	47	70	140	250
%MGI	7	37	47	63	75	79	79	82	84	100

**Table 3 ijms-25-01022-t003:** Inhibition of spore germination in *B. cinerea* via heptacyclosordariolone.

Inhibition of Spore Germination								
[ ] μg·mL^−1^	5	10	25	35	47	70	140	250
% inhibition	65	76	83	83	90	96	100	100

**Table 4 ijms-25-01022-t004:** Strains used in this study.

Strains	Species	Origin of Isolate	References
WT:B05.10	*B. cinerea*	*Vitis vinifera*	[63]
ST1-UCA	*S. tomento-alba*	*Gliricidia sepium*	This study

## Data Availability

The original contributions presented in the study are included in the article/Appendix A, further inquiries can be directed to the corresponding author/s. Sequences were submitted to NCBI database with the next accession number: OR835245 for ITS region; OR835246 for 28S rRNA gene; and OR879043 for β-Tubulin gene (https://www.ncbi.nlm.nih.gov/).

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
