# Peer review of "In Vitro Analysis of the Antagonistic Biological and Chemical Interactions between the Endophyte Sordaria tomento-alba and the Phytopathogen Botrytis cinerea"

_ijms, 2024, doi:10.3390/ijms25021022_

Round 1

Reviewer 1 Report

Comments and Suggestions for Authors

  This study isolated a fungus from a healthy leaf. It was identified based on molecular technique. It showed suppressive effects on Botrytis cinerea through production of antifungal metabolites (volatiles and non-volatiles). However, the authors did not show any data about efficacy of this fungus and its metabolites in suppression of gray mold disease. The authors should carefully make conclusions. 

Author Response

Dear Reviewer 1,

We sincerely appreciate your insightful comments and suggestions regarding our manuscript.

Our study primarily aimed to demonstrate the inhibitory potential of an endophytic fungus, isolated from the G. sepium plant, against B. cinerea in vitro. Additionally, we sought to evaluate the fungus's ability to produce bioactive molecules and to characterize these molecules during co-cultivation with B. cinerea. The results we presented focused on characterizing the newly isolated fungus and its bioactive compounds.

In alignment with your observation, our research team is currently investigating a suitable plant model susceptible to B. cinerea, where S. tomento alba can establish itself as an endophyte. We are also in the process of standardizing an optimal inoculation method for S. tomento alba to facilitate trials in greenhouse-inoculated plants. We value your suggestion regarding conducting plant-based tests; indeed, this forms a part of our ongoing research and future academic theses and publications. In response to your feedback, we have added a clarification in lines 652-656 of our manuscript (marked in red colour). This amendment explicitly states that our current findings do not conclusively determine the efficacy of this fungus and its metabolites in suppressing gray mold disease.

We hope you will find the revised version of our manuscript engaging and an improved reflection of our research intentions.

Yours sincerely,

Reviewer 2 Report

Comments and Suggestions for Authors

The paper of Bolívar-Anillo et al. entitled "In vitro analysis of the antagonistic biological and chemical interactions between the endophyte Sordaria tomento-alba and the phytopathogen Botrytis cinerea’ represents an interesting contribution in the characterization of new compounds with promising antifungal properties. The research was well conducted, and the paper is well written. Heptacyclosordariolone, one of the compounds isolated, caused inhibition of the mycelial growth as well as of conidia germination.

 The authors could add some more information on heptacyclosordariolone and structurally closely related compounds.

Heptacyclosordariolone was first characterized more than 2 decades ago by Bouillant et al., as indicated by the authors. Heptacyclosordariolone and heptacyclosordarianone in Sordaria sp. have been recently reported and characterized by Tsague et al. (2020). The authors omitted to discuss and contrast this finding in the discussion section. Also, compounds similar to heptacyclosordariolone (benzophomopsin A) have been identified in other endophytic fungi and their antimicrobial activity characterized (Shiono et al, 2009). Finally, the results could be discussed in the more general context of secondary metabolites from the Sordiariales that have antifungal and antibacterial activities (for example, Charria-Girón et al, 2022)

Figure 6, the formula of heptacyclosordariolone is deformed. Also, the same structure is presented in figure 7.

Is indeed Colombia ”the world's second most biodiverse country”? Any references supporting this assertion? Some websites provide different information, for example, https://news.mongabay.com/2016/05/top-10-biodiverse-countries.

Please check the references and make sure that species names are italicized. For example, references 54, 63, 68.

Tsague Tankeu VF, Sema DK, Jouda J-B, et al. Heptacyclosordarianone, a New Polyketide From Sordaria sp., an Endophytic Fungus From Garcinia polyantha. Natural Product Communications. 2020;15(12). doi:10.1177/1934578X20977622

Shiono, Y., Nitto, A., Shimanuki, K. et al. A new benzoxepin metabolite isolated from endophytic fungus Phomopsis sp.. J Antibiot 62, 533–535 (2009). https://doi.org/10.1038/ja.2009.65

Charria-Girón, E., Surup, F. & Marin-Felix, Y. Diversity of biologically active secondary metabolites in the ascomycete order SordarialesMycol Progress 21, 43 (2022). https://doi.org/10.1007/s11557-022-01775-3

Author Response

Dear Reviewer 2,

We would like to express our gratitude for your valuable comments on our manuscript.

We have carefully reviewed the articles you kindly suggested and have incorporated new information in the discussion (lines 417-422 and 431-433; in blue colour), along with the suggested references. Regarding other compounds isolated from Sordaria, we have added new information in the discussion (lines 417-422) about molecules whose antimicrobial activity has been evaluated. The remaining molecules described by Charria-Girón et al, 2022 for this genus have only been identified structurally, and while they have been reported to possess antioxidant and immunosuppressive capacities, their antimicrobial potential, which was the focus of our study, has not been included. We considered not include the reference Shiono et al., 2009, as it pertains to a different fungus, specifically Phomopsis sp., and though the compound structures are similar and structurally related, they display an organic function, an oxirane ring, different to those of heptacyclosordariolone, and it has not been assayed by its antimicrobial potential.

Furthermore, following the reviewer's suggestion, we have removed the chemical structure of heptacyclosordariolone from Figure 6, as it was indeed distorted, and the correct structure can be found in Figure 7.

In addition, in response to your suggestion, we have made a change in lines 659-660 (in blue colour) and included reference 69. The current version simply states that Colombia is one of the most biodiverse countries, acknowledging that such rankings vary depending on the variables considered.

Lastly, we have reviewed all the bibliographic references to ensure that the names of the genera and species of microorganisms and other organisms are italicized, along with correcting other minor typographical errors.

We are immensely grateful for the reviewer's comments, as they have significantly contributed to the improvement of our manuscript. We sincerely hope that this revised version meets your approval and that it is suitable for publication in the journal.

Yours sincerely,

Reviewer 3 Report

Comments and Suggestions for Authors

Interesting article on biocontrol against Botrytis cinerea. This is an important topic because so far the main role in combating mycelium is played by chemicals, which are not always beneficial to people and the environment. S. tomento-alba showed significant in vitro effects, inhibiting the growth of B. cinerea by approximately 60% when co-cultured and 50% when co-cultured on two discs. This is quite a promising result, however, more research is needed to be able to use S. tomento-alba as a biological agent to control B. cinerea.

Materials and methods as well as results are described in detail. References contain enough items. Figures described complement the obtained results. Statistical analysis was performed, statistically significant differences were found, demonstrating the influence of the tested organism on the pathogenic organism.

Author Response

Dear Reviewer 3,

We would like to express our gratitude for your comments on our manuscript.

We agree that the investigation into using S. tomento-alba as a biological agent to control B. cinerea is a highly promising area of research. We acknowledge that further research is essential to fully realize the potential of S. tomento-alba in this role.

We are immensely grateful for the reviewer's comments, and we sincerely hope that this revised version could be suitable for publication in the journal.

Yours sincerely,